# Novel View Synthesis from A Few Glimpses via Test-Time Natural Video Completion

Yan Xu[1]      Yixing Wang[1]      Stella X. Yu[1,2]

[1]University of Michigan     [2]UC Berkeley

{yxumich,yixingw,stellayu}@umich.edu

## Abstract

Given just a few glimpses of a scene, can you imagine the movie playing out as the camera glides through it? That's the lens we take on *sparse-input novel view synthesis*, not only as filling spatial gaps between widely spaced views, but also as *completing a natural video* unfolding through space.

We recast the task as *test-time natural video completion*, using powerful priors from *pretrained video diffusion models* to hallucinate plausible in-between views. Our *zero-shot, generation-guided* framework produces pseudo views at novel camera poses, modulated by an *uncertainty-aware mechanism* for spatial coherence. These synthesized frames densify supervision for *3D Gaussian Splatting* (3D-GS) for scene reconstruction, especially in under-observed regions. An iterative feedback loop lets 3D geometry and 2D view synthesis inform each other, improving both the scene reconstruction and the generated views.

The result is coherent, high-fidelity renderings from sparse inputs *without any scene-specific training or fine-tuning*. On LLFF, DTU, DL3DV, and MipNeRF-360, our method significantly outperforms strong 3D-GS baselines under extreme sparsity. Our project page is at https://decayale.github.io/project/SV2CGS.

## 1 Introduction

Humans can effortlessly imagine how a scene appears from unseen viewpoints by mentally filling in gaps, by drawing on prior visual experience to infer what's missing. Inspired by this ability, we reinterpret novel view synthesis – a long-standing challenge in computer vision and graphics [8, 21, 11, 42, 31, 66, 44, 18, 27] – as the task of completing a natural video from sparse camera views (Fig. 1). From this perspective, sparse-input novel view synthesis becomes analogous to recovering missing frames in a video captured along an unconstrained camera trajectory. This framing naturally invites the use of powerful generative priors learned from large-scale video data. In particular, pretrained video diffusion models [5, 55], which are trained to synthesize coherent and realistic scene motions, offer a compelling tool for filling in plausible scene content between widely spaced views.

Recently, NeRF [31, 2, 4, 32] and 3D Gaussian Splatting (3D-GS) [18, 61, 13, 15, 28] have significantly advanced novel view synthesis. Unlike NeRF, which represents scenes using an implicit function, 3D-GS models scenes explicitly with a set of 3D Gaussian primitives and renders images through efficient rasterization. 3D-GS achieves photorealistic rendering with substantially faster inference speed, making it a focal point of recent research interest.

However, synthesis from sparse inputs remains difficult. NeRF or 3D-GS methods typically rely on dense input views to accurately constrain the optimization process. In sparse-view settings, occlusions and geometric ambiguities [63] often lead to rendering artifacts and degraded quality. Recent efforts [22, 67, 10, 16, 50, 52] focus more on constrained camera paths (e.g., object-centric or forward-facing views). In contrast, real-world image capture from walking with a handheld smartphone often produces widely spaced, unconstrained views with large occlusions and out-of-view regions (Fig. 1).

39th Conference on Neural Information Processing Systems (NeurIPS 2025).

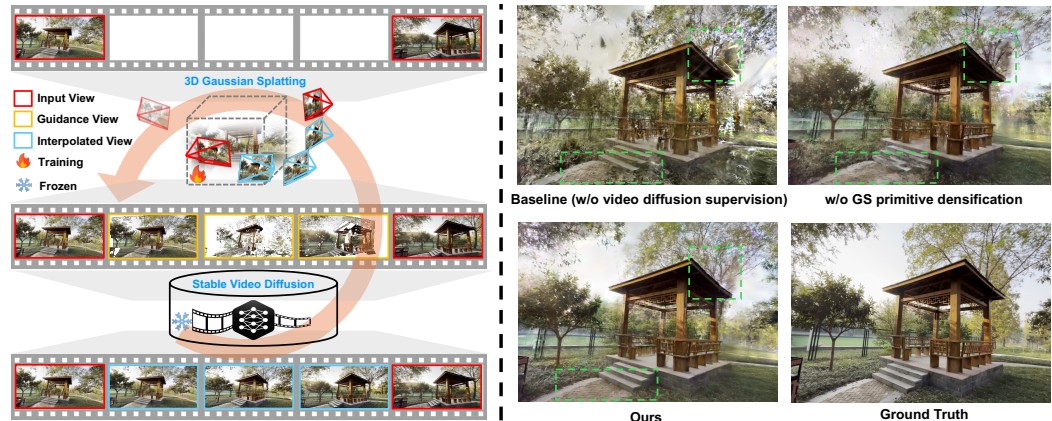

Figure 1: We view sparse-input novel view synthesis as temporal-spatial completion of a natural-looking video. **Left:** Our generation-guided reconstruction pipeline. With the initialized 3D-GS from sparse input views, ① we create guidance images on interpolated poses and estimate their uncertainty, based on the currently optimized 3D-GS. ② Using both guidance images and their uncertainties, we modulate the diffusion score function to interpolate between sparse input views. ③ The interpolated views are used to constrain 3D-GS optimization. **Right:** With our generation-guide reconstruction, the under-observed regions in the inputs are enhanced by the views generated by the diffusion model.

Motivated by the natural video completion perspective and strong priors in pretrained video diffusion models, we propose a **zero-shot, generation-guided reconstruction** pipeline integrating video diffusion with 3D-GS. Our approach defines target camera trajectories between sparse input views and uses video diffusion priors to synthesize plausible intermediate pseudo-views. These views provide supervision to better constrain 3D-GS training, especially in the under-observed regions in the inputs.

To recover missing views along a natural video trajectory, we must generate images at specified camera poses. However, existing video diffusion models [5, 6, 19, 45] are typically conditioned only on the initial frame and produce uncontrolled camera motions. While recent methods [49, 60] introduce trajectory conditioning during training, they still lack guarantees of pose alignment at inference and rely heavily on datasets with camera parameters, limiting generalization and scalability.

We propose a novel *uncertainty-aware modulation mechanism* that couples video diffusion with 3D Gaussian Splatting (3D-GS), enabling accurate, controllable frame interpolation under sparse-view settings. In this setup, 3D-GS provides a consistent 3D representation to guide view synthesis, while synthesized frames serve as pseudo supervision to further refine the 3D-GS model.

Fig. 1 illustrates our overall workflow. Our method begins by initializing 3D-GS from sparse views. After initialization, we interpolate camera poses between sparse inputs and create corresponding guidance images on the interpolated poses by inversely warping pixels from the nearest input view. The warping process is based on the depth maps rendered by the currently optimized 3D-GS. These guidance images are essential to maintaining the content and structural consistency during view interpolation, but may contain missing parts and artifacts due to imperfect 3D-GS depths and occlusion. We thus further model the uncertainty of these guidance images by assessing cross-view consistency in terms of photometry and geometry, and thereby focus the diffusion process more on correcting high-uncertainty regions, while keeping the reliable parts. Using both the guidance images and their associated uncertainties, we adaptively modulate the diffusion process to interpolate between the sparse views. The interpolated pseudo views are then added to the training set of 3D-GS. Furthermore, to improve the scene completeness for 3D-GS, we propose a *Gaussian primitive densification module* to densify the 3D-GS point cloud in under-observed regions using these pseudo views as bridges. The process above is repeated iteratively to refine the 3D-GS reconstruction.

To summarize, our contributions are threefold: **1)** We propose a *zero-shot, generation-guided* 3D-GS pipeline that leverages pretrained video diffusion models to improve novel view synthesis under sparse inputs. **2)** We introduce an *uncertainty-aware modulation* mechanism to integrate 3D-GS with video diffusion for controllable pseudo-view generation, and a *Gaussian primitive densification* module to enhance scene completeness. **3)** Our method achieves state-of-the-art performance, with

over 2.5 dB PSNR gain on DL3DV and strong results on LLFF and DTU, demonstrating robust generalization. While we primarily use Stable Video Diffusion [5], our framework is agnostic to the diffusion backbone and compatible with alternatives [55, 19].

## 2 Related Work

**Sparse-input Novel View Synthesis.** Sparse-input novel view synthesis aims to reconstruct a representation for generating novel views of a scene using a few input images. Although existing training-based methods, *i.e.* NeRF [31] and 3DGS [18], work well with dense inputs, their performance drops significantly with sparse views due to overfitting [37, 46, 33, 12, 39]. Several recent works explore robust novel view synthesis under sparse inputs. One group [7, 33, 16, 43, 40, 20, 67, 56] focuses on imposing additional regularization on views deviating from the training views. For example, GeoAug [7] randomly samples novel views around input frames and constrains rendering to match the input after view warping. Niemeyer *et al.* [33] introduce smooth depth regularization on unseen views. SPARF [43], GeCoNeRF [20], and FewViewGS [56] integrate multi-view correspondence and geometry loss into optimization. However, these methods do not address the fundamental issue of information deficiency in unobserved regions.

Another line of methods explores including priors from pre-trained neural networks [12, 46, 51, 34, 67, 22] for regularization. For example, Jain *et al.* [16] leverage CLIP [36] features to provide semantic guidance. DSNeRF [12] and SparseNeRF [46] use depth regularization from pre-trained depth estimators on known views to guide optimization. More recently, FSGS [67] and DNGaussian [22] extend the similar sprit to 3D-GS training. However, these priors do not directly provide visual supervision for sparse-view NVS like the visual diffusion prior.

**Novel View Synthesis with Diffusion Priors**. To leverage visual priors, several lines of work have emerged. Liu *et al.* [26] use diffusion models to generate pseudo-observations at unseen views, while Wu *et al.* [50] guide diffusion with a NeRF representation [58] to synthesize novel views.

To reduce the computational burden of fine-tuning diffusion models, Xiong *et al.* [52] and Wang *et al.* [47] adopt Score Distillation Sampling (SDS) [35] to extract external visual priors. However, these approaches rely on image-based diffusion models and thus fail to fully capture spatiotemporal correlations across views. More recently, Liu *et al.* [25] fine-tuned Stable Video Diffusion [5] to provide view interpolation capability for guiding 3D-GS reconstruction. While this significantly improves performance, it requires substantial computational resources, limiting practical efficiency.

Despite progress in view-conditioned generative models [24, 48, 38, 62], existing methods are either object-centric [24] or struggle to generate photorealistic views [38, 62, 48]. Recent approaches [14, 57, 49, 60] enable coarse camera motion control for video generation from a single frame but lack a consistent 3D representation, which compromises cross-view consistency and reproducibility.

Consequently, how to effectively leverage zero-shot video diffusion priors for novel view synthesis is an important open challenge. The concurrent work [65] is closely related to ours, but it depends on a video diffusion model trained with camera poses [60], and the code was not publicly available at the time of our submission. In contrast, our method can, in principle, be applied to any video diffusion model trained on raw videos, making it more broadly generalizable.

## 3 Preliminaries – More Details in Appendix

**3D Gaussian Splatting** (3D-GS) [18] represents 3D scenes explicitly using Gaussian primitives, each defined by mean $\boldsymbol{\mu} \in \mathbb{R}^3$ and covariance $\boldsymbol{\Sigma} \in \mathbb{R}^{3 \times 3}$: $G(\boldsymbol{x}) = \exp\left(-\frac{1}{2}(\boldsymbol{x} - \boldsymbol{\mu})^\top \boldsymbol{\Sigma}^{-1}(\boldsymbol{x} - \boldsymbol{\mu})\right)$. Each Gaussian also includes spherical harmonics coefficients $\boldsymbol{c}$ for view-dependent color and an opacity $\alpha$, enabling expressive appearance modeling. Rendering is performed efficiently via rasterization. After projecting Gaussians to the image plane, pixel colors are computed using alpha compositing: $C_{\text{pix}} = \sum_i \boldsymbol{c}_i \alpha_i \prod_{j=1}^{i-1}(1 - \alpha_j)$, where $\boldsymbol{c}_i$ and $\alpha_i$ denote the color and opacity of the $i$-th Gaussian, respectively. For depth rendering, $\boldsymbol{c}_i$ is replaced by the z-buffer value.

**Stable Video Diffusion** (SVD) [5] is an image-to-video diffusion model that generates natural video conditioned on an input image. By default, generation starts from the given image and autonomously evolves, incorporating random camera movements and scene dynamics.

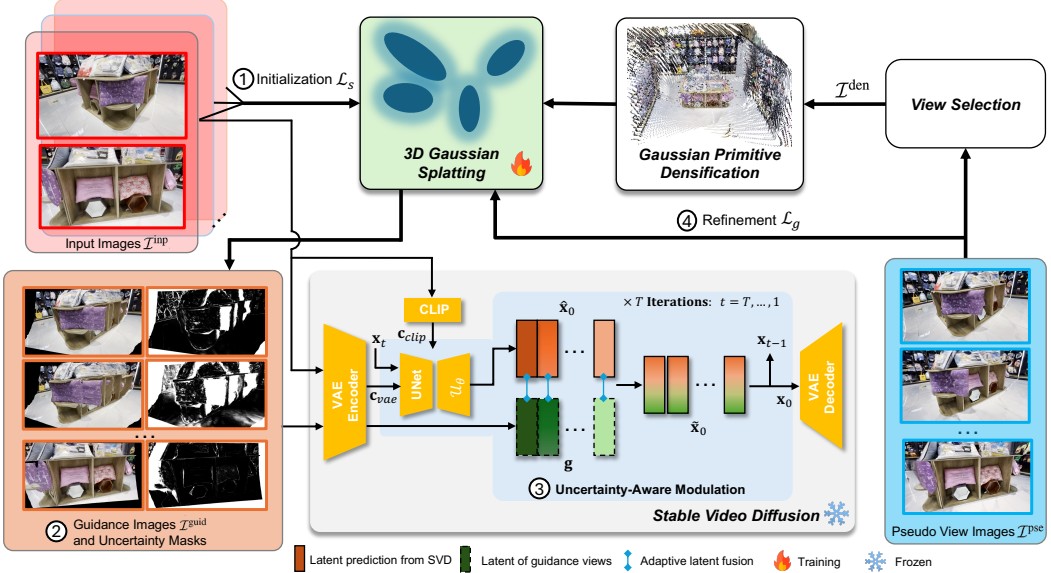

Figure 2: **Our approach leverages uncertainty-aware diffusion to synthesize pseudo views from sparse inputs and uses them to refine 3D Gaussian Splatting.** ① We initialize 3D-GS from sparse input images. ② We generate guidance images and estimate their uncertainties based on the current 3D-GS renderings. ③ These guidance images guide the diffusion process via uncertainty-aware modulation, enhancing uncertain regions while preserving reliable parts. ④ The resulting pseudo-view images are then used to densify Gaussian primitives and constrain the 3D-GS training.

Given a forward diffusion process expressed by $d\mathbf{x} = f(t)\mathbf{x}dt + g(t)d\mathbf{w}$, where $\mathbf{x}$ is the noisy latent state at timestamp $t$, $\mathbf{w}$ denotes the standard Wiener process, and $f(t)$ and $g(t)$ are scalar functions, its reverse process ODE [41] can be expressed as $d\mathbf{x} = \left[ f(t)\mathbf{x} - \frac{1}{2}g^2(\mathbf{x})\nabla_\mathbf{x}\log(q_t(\mathbf{x})) \right] dt$. In the case of the variance exploding (VE) diffusion [41] adopted by Stable Video Diffusion (SVD) [5], it can be simplified as: $d\mathbf{x} = \frac{\mathbf{x}-\hat{\mathbf{x}}_0}{\sigma_t}d\sigma_t$, where the noise of the diffusion process is parameterized as Gaussian noise with a variance of $\sigma_t$ and $\hat{\mathbf{x}}_0$ is the currently predicted clean video by the network based on the latent state at the previous step. In practice, we can obtain the estimated denoised sample $\mathbf{x}_{t-1}$ at the previous time step by discretizing the diffusion process above:

$$\mathbf{x}_{t-1} = \mathbf{x}_t + \frac{\mathbf{x}_t - \hat{\mathbf{x}}_0}{\sigma_t}(\sigma_{t-1} - \sigma_t). \tag{1}$$

## 4 Our Test-Time Optimization Approach to Novel View Synthesis

We recast sparse-input novel view synthesis as a test-time natural video completion problem. To this end, we propose an iterative optimization framework that integrates 3D Gaussian Splatting with video diffusion priors to enforce geometric consistency and enhance visual fidelity.

Given a few input views $\mathcal{I}^{inp}$ and their associated camera poses, we propose a zero-shot, generation-guided reconstruction pipeline that synthesizes novel views by leveraging a pretrained video diffusion model [5] (Fig. 2). The framework consists of four main steps: **1) 3D-GS initialization** from the sparse input views; **2) Guidance feature creation** and their **uncertainty estimation** via a cross-view consistency check based on the current 3D-GS; **3) Uncertainty-aware modulation** of the video diffusion model in generating pseudo views, conditioned on the guidance images and uncertainty masks; **4) Refinement of the 3D-GS** by densifying the Gaussian primitives using the generated pseudo-views. Steps **2)–4)** are iteratively performed to progressively improve both the 3D-GS representation and the quality of the diffusion model outputs.

### 4.1 Pseudo View Generation via Uncertainty-Aware Modulation

Most off-the-shelf video diffusion models lack precise camera control due to the scarcity of datasets with known camera poses. To ensure broad applicability, we design our framework to be compatible with widely available models [5, 55] that are conditioned solely on a single image. Moreover, our approach is theoretically agnostic to variance-exploding diffusion backbones [41].

Video diffusion models [5] usually extract CLIP [36] features $\mathbf{c}_{\text{clip}}$ from the input frame $I^{\text{inp}}$ to inform the U-Net of the scene's overall appearance and layout. Simultaneously, the frame is encoded by a VAE encoder to produce contextual features $\mathbf{c}_{\text{vae}}$, which are injected via classifier-free guidance to maintain consistency with the reference frame. At each denoising timestep $t$, the model denoises a latent video representation $\mathbf{x}_t \in \mathbb{R}^{N \times C \times H \times W}$ using a U-Net $\mathcal{U}_{\boldsymbol{\theta}}(\mathbf{x}_t; \mathbf{c}_{\text{clip}}, \mathbf{c}_{\text{vae}}, t)$, where $N$, $C$, $H$, $W$ are the number of frames, feature and spatial dimensions of the latent, respectively. The U-Net predicts a clean latent $\hat{\mathbf{x}}_0$ from $\mathbf{x}_t$ to update $\mathbf{x}_t$ with Eq. (1), which direct $\mathbf{x}_t$ toward $\hat{\mathbf{x}}_0$. The final denoised latent, $\mathbf{x}_0$, is decoded by the VAE decoder into a video clip.

Our method draws inspiration from diffusion-based image editing techniques [29, 59, 1, 53], particularly SDEdit [29] for its efficiency. Specifically, we propose to modify the original clean latent prediction $\hat{\mathbf{x}}_0$ using the guidance feature $\mathbf{g} \in \mathbb{R}^{N \times C \times H \times W}$ extracted from the guidance images by the VAE encoder. This modification is formulated as an optimization problem applied to each frame $i$:

$$\widetilde{\mathbf{x}}_0[i] = \arg\min_{\mathbf{x}} \|\mathbf{x} - \hat{\mathbf{x}}_0[i]\|_2^2 + \gamma_{t,i}\|\mathbf{x} - \mathbf{g}[i]\|_2^2, \qquad (2)$$

where index $[i]$ denotes the $i$-th frame channel corresponding to the $i$-th frame of the generated video, and $\gamma_{t,i} > 0$ is a weighting term that controls the influence of the guidance feature. Next, we describe how to obtain the feature map $\mathbf{g}$ that guides the diffusion model to generate views at desired poses, and detail how to control $\gamma_{t,i}$ to achieve adaptive modulation.

**Guidance Feature Creation.** The core idea of our approach is to exploit video diffusion priors to infer occluded or missing content from sparse input views. This requires constructing guidance features that are geometrically aligned with the desired target view.

To resolve this, instead of using the 3D-GS to render color images, we create guidance images by inversely warping pixels from their nearest input view, using depth maps rendered by 3D-GS. Concretely, to construct the guidance image $I_i^{\text{guid}}$ for the $i$-th video frame, we first project each pixel $\mathbf{p} \in I^{\text{guid}}$ into the nearest input view $I^{\text{inp}} \in \mathcal{I}^{\text{inp}}$, using the rendered depth map $D_i^{\text{guid}}$, camera intrinsics $\mathbf{K}$, and camera poses $\mathbf{P}^{\text{inp}} \in \mathbb{SE}(3)$ (input view) and $\mathbf{P}_i^{\text{guid}} \in \mathbb{SE}(3)$ (guidance view), to get its corresponding pixel $\mathbf{q}$ in the input image:

$$\mathbf{q} = \mathbf{K}\mathbf{P}^{\text{inp}}(\mathbf{P}_i^{\text{guid}})^{-1} D_i^{\text{guid}}(\mathbf{p})\mathbf{K}^{-1}\mathbf{p}. \qquad (3)$$

We fill pixel $\mathbf{p}$ with the color of pixel $\mathbf{q}$ to obtain the guidance image $I_i^{\text{guid}}$. The set of guidance images is denoted as $\mathcal{I}^{\text{guid}} = \{I_i^{\text{guid}}\}_{i=1}^N$, where $N$ is the length of the video clip generated by the video diffusion model in a single pass. The VAE encoder will encode these guidance images to have the corresponding guidance feature maps $\mathbf{g}$ to guide the diffusion process via Eq. (2).

**Uncertainty Evaluation from Cross-View Consistency.** The constructed guidance images well preserve scene content and structure by adhering to strict multi-view geometric constraints imposed by the 3D-GS representation. However, because 3D-GS is imperfect during training, especially in under-observed regions, the guidance images may contain missing content or artifacts. To assess the reliability of guidance images, we introduce a strict cyclic consistency check, as illustrated in Fig. 3a. Specifically, in the forward pass, we project each pixel $\mathbf{p}$ in the guidance image to its corresponding pixel $\mathbf{q}$ in the nearest input image using Eq. (3). We then perform a backward projection from $\mathbf{q}$ to the guidance view using the depth map $D^{\text{inp}}$ rendered by 3D-GS from the nearest input view: $\mathbf{p}' = \mathbf{K}\mathbf{P}_i^{\text{guid}}(\mathbf{P}^{\text{inp}})^{-1}D^{\text{inp}}(\mathbf{q})\mathbf{K}^{-1}\mathbf{q}$. The uncertainty at pixel $\mathbf{p}$ is then quantified by evaluating both geometric and photometric consistency:

$$U_i(\mathbf{p}) = 1 - \exp\left(-\frac{||\mathbf{p} - \mathbf{p}'||_2^2}{s_1} - \frac{||I_i^{\text{gs}}(\mathbf{p}) - I^{\text{inp}}(\mathbf{q})||_2^2}{s_2}\right), \qquad (4)$$

where $I_i^{\text{gs}}$ is the 3D-GS rendered image from the view of the $i$-th guidance image, $I^{\text{inp}}$ denotes the nearest input image, and $s_1$, $s_2$ are bandwidth parameters controlling the sensitivity to geometric and

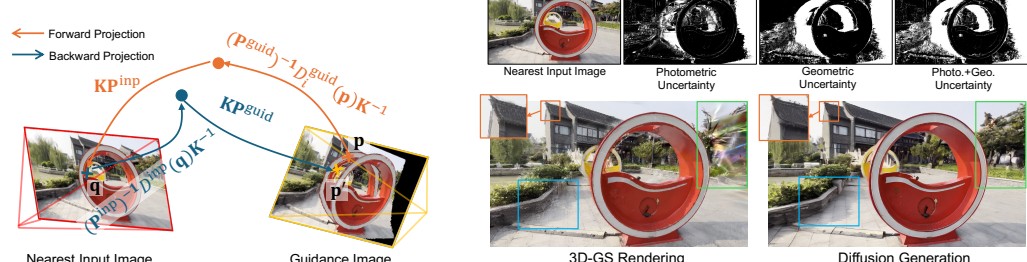

(a) Illustration of Forward/Backward Projection      (b) Example of the Uncertainty Estimates

Figure 3: **Cross-view consistency provides a principled uncertainty estimate for guidance images, enabling targeted refinement by video diffusion.** (a) We evaluate cross-view consistency via a forward–backward projection cycle between a guidance view and its nearest input view using depths rendered from the current 3D-GS. (b) Regions exhibiting poor cross-view consistency (boxed regions in the rendering) are identified as high-uncertainty areas (brighter in uncertainty maps), and subsequently refined by the video diffusion model.

photometric discrepancies. If the 3D-GS is well constrained at pixel $\mathbf{p}$ and no occlusion is present, the image pixel color $I^{\text{inp}}(\mathbf{q})$ should closely match the color of the 3D-GS rendering $I_i^{\text{gs}}(\mathbf{p})$, and the back-projected position $\mathbf{p}'$ should lie near the original $\mathbf{p}$. This results in low uncertainty. Otherwise, discrepancies in color or geometry increase the uncertainty, as captured by Eq. (4).

**Uncertainty-Aware Modulation.** Using the uncertainty map, we define $\gamma_{t,i}$ for each pixel in Eq. (2):

$$\gamma_{t,i}(\mathbf{p}) = \begin{cases} 0 & U_i(\mathbf{p}) > \delta \text{ or } t < \tau \\ 1/(U_i(\mathbf{p}) + \epsilon) & \text{otherwise} \end{cases}, \tag{5}$$

where $\delta$ and $\tau$ are threshold hyperparameters and $\epsilon$ is a small constant to avoid division by zero. The threshold $\tau$ is determined by the overall uncertainty of frame $i$, defined by $\tau = \mathcal{F}(\frac{1}{HW}\sum_{\mathbf{p}}(U_i(\mathbf{p}))$ where $\mathcal{F}(\cdot)$ can be defined as a function (linearly or quadratically) increasing between 0 and 1 (see supplementary materials for details). This ensures that in uncertain regions, the optimization in Eq. (2) leans towards the diffusion prediction $\hat{\mathbf{x}}_0[i]$, while reliable areas are guided by the features from $\mathbf{g}[i]$. For simplicity, we let $\mathbf{p}$ denote corresponding positions in both image and latent space. In practice, $U_i$ is downsampled via average pooling to match the latent resolution before computing $\gamma_{t,i}$. After computing $\gamma_{t,i}$, we apply Eq. (2) to obtain the fused latent $\tilde{\mathbf{x}}_0[i]$, which is then used in Eq. (1) to update $\mathbf{x}_t$ to $\mathbf{x}_{t-1}$. This reverse sampling step is repeated until the final latent $\mathbf{x}_0$ is obtained, which is then decoded into pseudo-view images via the VAE decoder (see Fig. 2).

**Extending to View Interpolation.** The above pipeline supports single-view extrapolation but degrades under large viewpoint changes. We extend it to two-view interpolation by defining camera paths between inputs and running diffusion forward and backward, conditioned on each view. At each denoising step, the two latent sequences are merged: $\mathbf{x}_{t-1} := \boldsymbol{\beta}\mathbf{x}_{t-1}^{\text{forward}} + (1 - \boldsymbol{\beta})R(\mathbf{x}_{t-1}^{\text{backward}})$, where $R(\cdot)$ is the reverse operation along the frame index dimension to align the latent $\mathbf{x}_{t-1}^{\text{backward}}$ to $\mathbf{x}_{t-1}^{\text{forward}}$ in the frame dimension. $\boldsymbol{\beta} \in \mathbb{R}^N$ is the blending weight, with $\boldsymbol{\beta}[i] = (N - i)/(N - 1)$ for $i = 1, 2, \ldots, N$, where $N$ is number of interpolated frames between two inputs. See supplementary material for the detailed algorithm.

### 4.2 3D-GS Optimization Guided by Generation

To better constrain the 3D-GS representation, we pair adjacent inputs and define camera trajectories that cover under-observed regions (see supplement). Using the video diffusion model guided by $\mathcal{I}^{\text{guid}}$ (Sec. 4.1), we interpolate between input views to generate pseudo-view images $\mathcal{I}^{\text{pse}} = \{I_j^{\text{pse}}\}_{j=1}^{pN}$, where $p$ is the number of input pairs.

**Gaussian Primitive Densification.** Sparse-input 3D-GS often yields poor geometry in under-observed regions due to limited supervision. We address this by enhancing the geometry using pseudo-views $\mathcal{I}^{\text{pse}}$ and a dense stereo model [48]. For efficiency, we select a subset $\mathcal{I}^{\text{den}} \subseteq \mathcal{I}^{\text{pse}}$ with low inter-frame covisibility to maximize coverage with minimal redundancy. These views form a camera graph

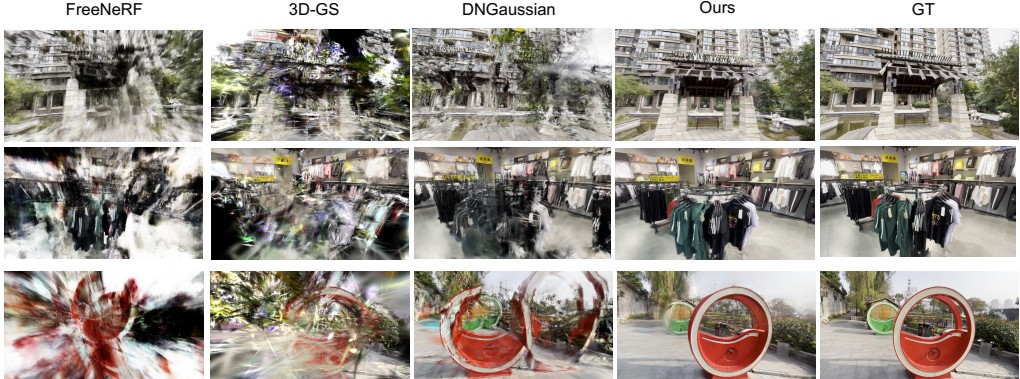

Figure 4: **Our generation-guided reconstruction produces more coherent and photorealistic novel views than prior methods.** We compare our results on the DL3DV dataset using 9 input views with those from FreeNeRF, vanilla 3D Gaussian Splatting, and DNGaussian. By leveraging video diffusion priors to complete under-observed regions, our method better preserves scene structure and appearance consistency.

used to reconstruct a point cloud from stereo predictions. We analyze the spatial distribution of reconstructed points, and filter out outliers that significantly deviate from the global average distance to neighboring points. We query existing Gaussian primitives within a radius of each remaining point and only add new Gaussian primitives at positions without nearby primitives to augment the current set. See appendix for more details.

**3D Gaussian Splatting Optimization.** After densifying the Gaussian primitive set, we optimize the 3D-GS model using both the original inputs and the generated pseudo-views. In each training iteration, one input view and one pseudo-view are sampled for supervision. For the original input views, we apply an L1 loss and a D-SSIM loss , as well as a depth regularization term $\mathcal{L}_{\text{reg}}$ with Pearson correlation similar to [47]: $\mathcal{L}_s = w_1\mathcal{L}_1(I^{\text{gs}}, I^{\text{inp}}) + w_2\mathcal{L}_{\text{D-SSIM}}(I^{\text{gs}}, I^{\text{inp}}) + w_3\mathcal{L}_{\text{reg}}$, where $I^{\text{gs}}$ is the rendered image from 3D-GS and $I^{\text{inp}}$ denotes the corresponding input image. For the generated pseudo views, we observe that, despite the carefully designed guidance mechanism, some regions still suffer from temporal inconsistency—particularly distant areas with weak geometry or those with fine-grained textures, *e.g.*, grass or tree leaves. To mitigate the negative impact of such inconsistencies on 3D-GS training, we use the LPIPS [64] instead of L1 loss. The resulting loss is:

$$\mathcal{L}_g = w_4\mathcal{L}_{\text{LPIPS}}(I^{\text{gs}}, I^{\text{pse}}) + w_5\mathcal{L}_{\text{D-SSIM}}(I^{\text{gs}}, I^{\text{pse}}) + w_6\mathcal{L}_{\text{reg}}. \qquad (6)$$

## 5 Experiments

### 5.1 Experiment Settings

**Datasets and Metrics**. We evaluate our method on LLFF [30], DL3DV [23], DTU [17], and MipNeRF-360 [3] datasets. LLFF consists of 8 forward-facing scenes. Following standard practice [54, 22], we train our model using only 3 input views on this dataset. DL3DV comprises diverse indoor and outdoor scenes captured by humans walking through scenes, exhibiting complex and dynamic camera motions. The Mip-NeRF 360 dataset consists of real-world indoor and outdoor scenes designed for evaluating novel view synthesis in large, unbounded environments. To verify the generalizability of our methods and compare with the previous methods, we also test our methods on DTU, an object-centric dataset captured in controlled conditions. For the DTU dataset, we follow the protocol from RegNeRF [22], using 3 training views across 15 evaluation scenes. To focus on the object of interest, we mask out the background during evaluation using the provided object masks, consistent with [54, 22]. We apply a downsampling factor of 8 for LLFF and 4 for DTU, aligning with prior work. The rendering quality is assessed using PSNR, SSIM, and LPIPS metrics.

**Implementation details**. Our pipeline is designed to operate iteratively. In each cycle, we train the 3D-GS model for 10K iterations, followed by an update of the pseudo-view images using the video

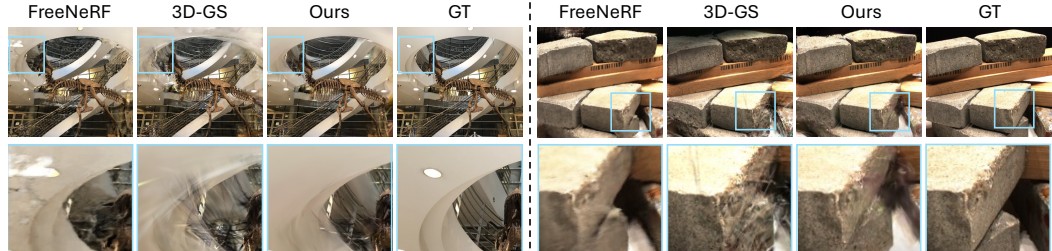

Figure 5: **Our method more faithfully recovers scene geometry and fine-grained appearance under sparse inputs.** We show comparisons on the LLFF (left) and DTU (right) datasets. Diffusion-generated pseudo views densify supervision in under-observed areas (boxed), enabling better preservation of structural integrity and high-frequency details.

Table 1: **Our method consistently outperforms prior approaches under sparse-view settings across multiple datasets.** We show comparisons on LLFF, DTU, and DL3DV. We compare our method with standard reconstruction baselines (vanilla NeRF / 3D-GS), sparse-view–optimized NeRF methods, and sparse-view–optimized 3D-GS methods. We color each cell as **best**, **second best**, and **third best**.

| | LLFF (3 Views) | | | DTU (3 Views) | | | DL3DV (3 Views) | | | DL3DV (6 Views) | | | DL3DV (9 Views) | | |
|---|---|---|---|---|---|---|---|---|---|---|---|---|---|---|---|
| | PSNR↑ | SSIM↑ | LPIPS↓ | PSNR↑ | SSIM↑ | LPIPS↓ | PSNR↑ | SSIM↑ | LPIPS↓ | PSNR↑ | SSIM↑ | LPIPS↓ | PSNR↑ | SSIM↑ | LPIPS↓ |
| Mip-NeRF | 16.11 | 0.401 | 0.460 | 8.68 | 0.571 | 0.353 | 10.92 | 0.191 | 0.618 | 11.56 | 0.199 | 0.608 | 12.42 | 0.218 | 0.600 |
| 3D-GS | 17.43 | 0.522 | 0.321 | 10.99 | 0.585 | 0.313 | 10.97 | 0.248 | 0.567 | 12.34 | 0.332 | 0.598 | 12.99 | 0.403 | 0.546 |
| DietNeRF | 14.94 | 0.370 | 0.496 | 11.85 | 0.633 | 0.314 | – | – | – | – | – | – | – | – | – |
| RegNeRF | 19.08 | 0.587 | 0.336 | 18.89 | 0.745 | 0.190 | 11.46 | 0.214 | 0.600 | 12.69 | 0.236 | 0.579 | 12.33 | 0.219 | 0.598 |
| FreeNeRF | 19.63 | 0.612 | 0.308 | 19.92 | 0.787 | 0.182 | 10.91 | 0.211 | 0.595 | 12.13 | 0.230 | 0.576 | 12.85 | 0.241 | 0.573 |
| SparseNeRF | 19.86 | 0.624 | 0.328 | 19.55 | 0.769 | 0.201 | – | – | – | – | – | – | – | – | – |
| SparseGS | – | – | – | 18.89 | 0.834 | 0.178 | – | – | – | – | – | – | – | – | – |
| FSGS | 20.31 | 0.652 | 0.288 | – | – | – | 12.22 | 0.296 | 0.535 | 13.73 | 0.429 | 0.540 | 15.52 | 0.468 | 0.416 |
| DNGaussian | 19.12 | 0.591 | 0.294 | 18.91 | 0.790 | 0.176 | 11.10 | 0.273 | 0.579 | 12.67 | 0.329 | 0.547 | 13.44 | 0.365 | 0.539 |
| IPSM | 20.44 | 0.702 | 0.207 | – | – | – | 11.70 | 0.279 | 0.534 | 12.82 | 0.332 | 0.521 | 13.41 | 0.361 | 0.529 |
| **Ours** | 20.61 | 0.705 | 0.201 | 20.51 | 0.840 | 0.137 | 14.62 | 0.471 | 0.491 | 17.35 | 0.566 | 0.396 | 19.19 | 0.616 | 0.335 |

diffusion model. After each pseudo-view update, we reset the learning rate schedule of 3D-GS before starting the next optimization cycle to avoid overfitting. For the uncertainty estimation in Eq. (4), we set the bandwidth parameters to $s_1 = 100$ and $s_2 = 0.25$. The $\delta$ in Eq. (5) is fixed at $0.5$ across all experiments. The loss weights are configured as follows: $w_1 = 0.8$, $w_2 = 0.2$, $w_3 = 1.0$, $w_4 = 1.0$, $w_5 = 0.2$, and $w_6 = 1.0$. Additional implementation details are provided in supplementary materials.

## 5.2 Comparison with Other Methods

We compare our method against the state-of-the-art on four benchmark datasets to demonstrate its effectiveness and generalizability across diverse scenarios.

**Comparison on LLFF**. We evaluate our method on the LLFF dataset captured by a swaying face-forward camera. Table 1 shows that our method consistently outperforms NeRF-based approaches across all evaluation metrics. When compared to 3D-Gaussian Splatting–based baselines such as FSGS [67] and DNGaussian [22], our method remains competitive, particularly in LPIPS and SSIM scores. This improvement is largely attributed to the additional supervisory signal provided by the pseudo views generated through the video diffusion model. Notably, the LPIPS metric, which correlates more closely with human perceptual similarity than traditional metrics like PSNR, highlights our method's ability to produce visually realistic novel views. Qualitative comparisons are presented in Fig. 5.

Table 2: **Our method outperforms recent diffusion-based feed-forward approaches.** Results are reported on MipNeRF-360 using 9 input views. Our approach surpasses feed-forward approaches (e.g. MVSplat 360) without expensive diffusion finetuning.

| *MipNeRF-360* | PSNR↑ | SSIM↑ | LPIPS↓ |
|---|---|---|---|
| RegNeRF [33] | 13.73 | 0.193 | 0.629 |
| FreeNeRF [54] | 13.20 | 0.198 | 0.635 |
| DNGaussian [22] | 12.51 | 0.228 | 0.683 |
| MVSplat 360 [9] | 14.86 | 0.321 | 0.528 |
| ViewCrafter [60] | 16.68 | 0.382 | 0.551 |
| 3DGS-Enhancer [25] | 16.22 | 0.399 | 0.454 |
| **Ours** | 17.91 | 0.495 | 0.435 |

**Comparison on DTU**. To further assess the generalizability of our approach, we evaluate and compare its performance on the DTU dataset. DTU is an object-centric dataset in which each scene

Table 3: **Ablation experiments on the DL3DV test set**. (a) Experiments to show the effectiveness of the proposed components in pseudo-view generation step. (b) Experiments to show the effectiveness of the proposed strategies for 3D-GS optimization.

| (a) Pseudo-view generation | PSNR↑ | SSIM↑ | LPIPS↓ |
|---|---|---|---|
| Baseline 3D-GS | 16.59 | 0.502 | 0.405 |
| w/ GS interpolation | 18.59 | 0.591 | 0.369 |
| w/ warping interpolation (full) | **19.19** | **0.616** | **0.335** |
| w/o geometric | 18.21 | 0.583 | 0.378 |
| w/o photometric | 18.93 | 0.612 | 0.352 |

| (b) 3D-GS optimization | PSNR↑ | SSIM↑ | LPIPS↓ |
|---|---|---|---|
| w/o point filtering | 19.01 | 0.615 | 0.343 |
| w/o GS densification | 18.23 | 0.567 | 0.386 |
| w/o LPIPS loss | 18.81 | 0.597 | 0.351 |
| Full model | **19.19** | **0.616** | **0.335** |

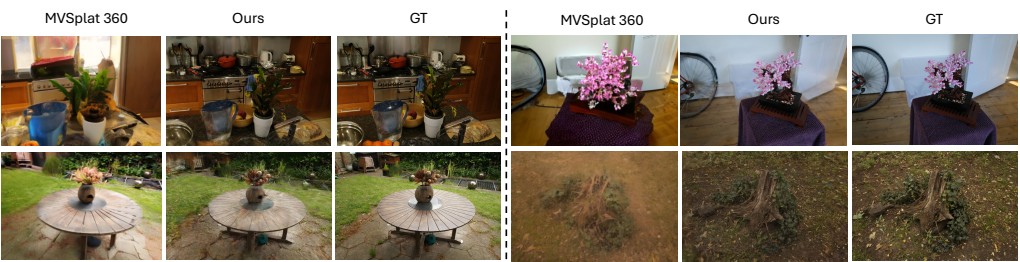

Figure 6: **Our test-time optimization better preserves visual and geometric consistency than the feed-forward approach, MVSplat360**. 9 views are taken as the input. While feed-forward methods can produce plausible novel views, they often struggle to maintain fidelity to the original scene, whereas our method achieves higher consistency.

contains a centered object against a monotone background. The evaluation results are presented in Table 1 (middle). In this setting, our method still performs well and outperforms other NeRF-based and 3D-GS-based methods. Specifically, our method outperforms the second-best approach by a significant margin in terms of PSNR, SSIM, and LPIPS. While NeRF-based methods also exhibit competitive accuracy in this scenario, they suffer from slow rendering speeds (approximately 0.21 FPS), whereas our 3D-GS-based approach supports real-time rendering at around 430 FPS.

**Comparison on DL3DV**. We compare with other cutting-edge counterparts on the DL3DV dataset under 3, 6, and 9 view settings. Table 1 (right) shows the quantitative comparison results. Apart from the sparse-input 3D-GS methods, we also compare with the non-sparse view methods and NeRF-based methods in Table 1 (right). We outperform previous state-of-the-art methods [22, 67, 47] by a significant margin in this challenging setting. We observe that although DNGaussian [22] works well in environments with limited scope or with limited camera motions, *e.g.*, object-centric scenarios, it has difficulties in reliably reconstructing the open environment due to the lack of constraints in under-observed regions (qualitative results shown in Fig. 4). Similarly, FSGS [67] also struggles in this challenging setting, though it achieves slightly better performance compared with DNGaussian because it uses a sparse point cloud for initialization. The recent work IPSM [47] uses an image diffusion model to constrain the 3D-GS by enhancing Score Distillation Sampling (SDS). As shown in Table 1 (right), this method struggles with extremely sparse inputs. This limitation arises because the image diffusion model lacks access to a global scene context, whereas the video diffusion model is able to infer such context from the input reference frame.

**Comparison on MipNeRF-360**. To evaluate our method on unbounded scenes and ensure a fair comparison with recent feed-forward approaches [9, 60, 25], we further conduct experiments on the Mip-NeRF 360 dataset [3]. As shown in Table 2, our method consistently outperforms reconstruction-based methods [33, 54, 22] and surpasses state-of-the-art feed-forward approaches [9, 60, 25] by a notable margin. As shown in Fig. 6, although feed-forward methods can hallucinate novel views from sparse inputs through large-scale data training, they often struggle to maintain geometric consistency, fine details, and color fidelity compared to our approach.

## 5.3 Ablation Study

To validate the effectiveness of our proposed components in the pseudo-view generation (Sec. 4.1) and the 3D-GS optimization (Sec. 4.2), we conduct an extensive ablation study on DL3DV.

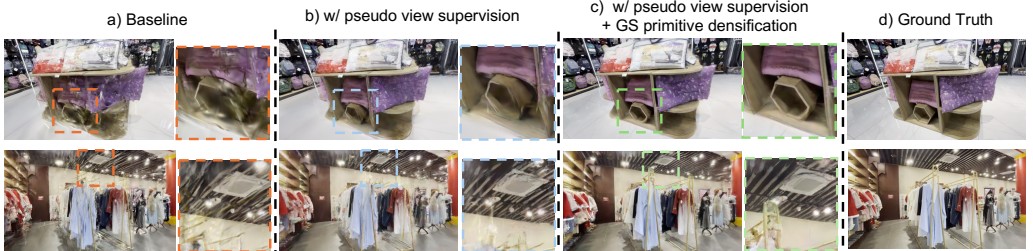

Figure 7: **The proposed pseudo-view supervision and primitive densification effectively enhance novel view synthesis.** We show two scenes with under-observed regions highlighted in boxes, comparing results with and without these components. Both strategies improve geometric reconstruction and visual fidelity in under-observed regions.

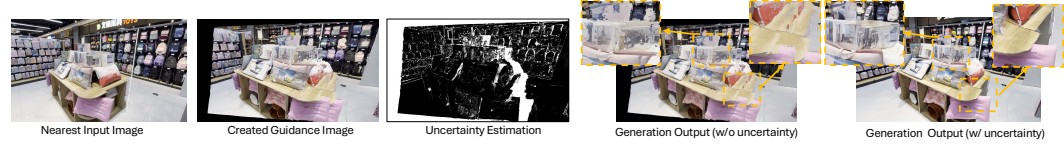

Figure 8: **Uncertainty-aware modulation is critical for reliable pseudo-view generation.** We compare diffusion outputs with and without uncertainty modulation, especially in unreliable regions (boxed). Uncertainty modeling effectively prevents artifacts.

**Effectiveness of uncertainty-aware modulation mechanism.** Table 3a compares the baseline 3D-GS trained on sparse views using $\mathcal{L}_s$ with two variants: one using 3D-GS renderings as guidance ("w/ GS interpolation") and one using our warping-based guidance ("w/ warping interpolation"). While GS interpolation improves over the baseline, it underperforms compared to our method due to inaccurate color rendering at novel poses during training.

Fig. 8 shows the effect of uncertainty-aware modulation by comparing diffusion results with and without it, using identical guidance images. We further ablate the geometric and photometric terms in the uncertainty formulation (Eq. (4)), denoted as "w/o geometric" and "w/o photometric." As shown in Table 3a, removing either term noticeably degrades performance.

**Effectiveness of Gaussian primitive densification.** We ablate the densification step ("w/o GS densification" in Table 3b), observing a significant performance drop, highlighting its role in improving synthesis quality. Fig. 7 shows that densification enhances reconstruction in under-observed regions. Removing the point filtering step ("w/o point filtering") also degrades performance due to depth outliers from the stereo model.

**Effectiveness of LPIPS for pseudo view supervision.** We replace LPIPS with L1 loss ("w/o LPIPS loss") in Eq. (6), observing a notable performance drop (Table 3b). Despite our guidance strategy, cross-view inconsistencies – especially in distant or textured regions – remain challenging. L1 loss used in vanilla 3D-GS [18] is less robust to such inconsistencies in diffusion-generated pseudo views.

## 6    Conclusion and Limitations

We present a zero-shot, generation-guided pipeline that leverages a pretrained video diffusion model to improve 3D-GS reconstruction from sparse inputs. The method synthesizes intermediate views guided by warped depth-based images and uncertainty-aware modulation, while a densification module further improves scene completeness. Our approach enhances photorealism and coverage in sparse settings while preserving the real-time efficiency of 3D-GS.

Despite these gains, the framework has limitations. Its performance depends on the quality of the pretrained video diffusion model, which may introduce artifacts under extreme viewpoints or in complex scenes. The iterative training procedure adds overhead relative to vanilla 3D-GS pipelines, and early-stage 3D-GS depth errors can affect the guidance despite uncertainty modeling, although this impact typically diminishes over time.

**Acknowledgments.** This project was supported, in part, by NSF 2215542, NSF 2313151, and Bosch gift funds to S. Yu at UC Berkeley and the University of Michigan, with additional compute support provided by the NAIRR Pilot under CIS240421.

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
