# OpenReview forum: "Novel View Synthesis from A Few Glimpses via Test-Time Natural Video Completion"
_NeurIPS.cc/2025/Conference — NeurIPS 2025 poster_

### Official Review · Reviewer_pPRG · 2025-06-27

**Clarity:** 3
**Significance:** 2
**Originality:** 2
**Rating:** 4
**Confidence:** 5

**Summary:**

The paper presents a zero-shot, generation-guided reconstruction framework that leverages pretrained video diffusion models to enhance 3D Gaussian Splatting (3D-GS) for sparse-input novel view synthesis. It reinterprets the task as natural video completion, using diffusion to generate pseudo views that provide dense supervision for 3D-GS training. An uncertainty-aware modulation mechanism and Gaussian primitive densification module are introduced to improve spatial coherence and scene completeness. Experiments on DL3DV, LLFF, and DTU show performance gains over baselines, but concerns remain regarding novelty, baseline comparisons, and visualization details.

**Questions:**

Please see the weaknesses.

**Ethical Concerns:**

["NO or VERY MINOR ethics concerns only"]

**Final Justification:**

The rebuttal well addressed my concerns on the comparisons with recent works. The authors are encouraged to demonstrate this clearly in the final version. I will raise my final score.

**Limitations:**

As described in the weaknesses, the major limitation of this paper is missing comparisons with the relevant ideas. From my point of view, a detailed comparison with these methods could further clarify the true contributions of this paper.

**Quality:**

2

**Strengths And Weaknesses:**

Strengths:

1. The uncertainty-aware modulation mechanism effectively guides diffusion for consistent view synthesis.

2. The iterative refinement between 3D-GS and diffusion models enhances scene reconstruction in under-observed regions.

Weaknesses:

1. Lack of novelty: Zero-shot, generation-guided 3D reconstruction has been previously explored in works like ReconX [1], ViewCrafter [2], Wonderland [3], See3D [4], VideoScene [5], and Difix3D+ [6]. The proposed framework does not sufficiently distinguish itself from these prior art methods.

2. More baseline comparisons: The paper claims state-of-the-art performance but omits recent advances such as those referenced in [1]-[6]. Comparing against the latest methods is essential to validate superiority. Additionally, since the approach relies on view interpolation, comparisons with two-view reconstruction methods (e.g., pixelSplat [7], MVSplat [8], NoPosSplat [9]) are suggested to be included into experiment.

3. Missing input condition visualization: Visualization experiments in the main paper lack explicit input sparse views. Including visualization of the sparse input images would better illustrate the method’s interpolation capabilities.

[1] ReconX: Reconstruct Any Scene from Sparse Views with Video Diffusion Model

[2] ViewCrafter: Taming Video Diffusion Models for High-fidelity Novel View Synthesis

[3] Wonderland: Navigating 3D Scenes from a Single Image

[4] You See it, You Got it: Learning 3D Creation on Pose-Free Videos at Scale

[5] VideoScene: Distilling Video Diffusion Model to Generate 3D Scenes in One Step

[6] Difix3D+: Improving 3D Reconstructions with Single-Step Diffusion Models

[7] pixelSplat: 3D Gaussian Splats from Image Pairs for Scalable Generalizable 3D Reconstruction

[8] MVSplat: Efficient 3D Gaussian Splatting from Sparse Multi-View Images

[9] No Pose, No Problem: Surprisingly Simple 3D Gaussian Splats from Sparse Unposed Images

---

> ### Author Rebuttal · Authors · 2025-07-30
>
> # Reviewer pPRG
>
> **Q1:** "Lack of novelty: Zero-shot, generation-guided 3D reconstruction has been previously explored in works like ReconX [1], ViewCrafter [2], Wonderland [3], See3D [4], VideoScene [5], and Difix3D+ [6]. The proposed framework does not sufficiently distinguish itself from these prior art methods."
>
> **A1:** We thank the reviewer for highlighting recent works such as ReconX [1], ViewCrafter [2], and other concurrent CVPR 2025 papers.
>
> Given that many of these methods involve 3D reconstruction and diffusion models, we understand the potential for confusion. However, our framework is *fundamentally distinct in two key ways*:
> 1. It is truly zero-shot — frozen SVD only, no model parameters trained at any stage, and
> 2. It is truly test-time — leveraging a pretrained video diffusion model as is without any task-specific training for novel view synthesis.
>
> In our manuscript, "zero-shot" refers specifically to using a frozen video diffusion model—trained for generic video generation—as a prior to guide 3D Gaussian Splatting (3D-GS) optimization for novel view synthesis. We do so without any fine-tuning or additional training, making our pipeline lightweight, modular, and broadly deployable.
>
> While the cited methods also leverage diffusion models, most of them **train those models explicitly for 3D or novel view synthesis**, using domain-specific datasets and large-scale compute. For example, See3D trains on **114 × A100 GPUs for ~25 days**. These models **are not plug-and-play**: they require **substantial pretraining tailored to the task**, and typically operate only within their trained data domain. In contrast, our method uses **off-the-shelf, frozen diffusion models** with **no retraining or adaptation**, enabling broader generalization and ease of use in real-world, low-resource scenarios.
>
> Furthermore, our focus is not on training a diffusion model to regress 3D scenes or render views directly, but on **enhancing 3D-GS reconstruction quality under sparse-view constraints**, while **retaining its real-time rendering advantages** (~400 FPS). Our pipeline also introduces a key technical innovation— uncertainty-aware modulation —that steers the frozen diffusion model to synthesize pseudo-views between sparse inputs, which are then used to supervise 3D-GS in under-observed regions.
>
> This modulation is part of an **iterative feedback loop** that jointly refines both the 3D representation and synthesized views, yielding significant gains in fidelity and consistency (see Fig. S-3). We are not aware of any prior work that combines frozen diffusion priors with 3D Gaussian Splatting in this manner.
>
> While approaches like ReconX or ViewCrafter could hypothetically be paired with 3D-GS, they lack **test-time modulation, iterative supervision, and zero-shot adaptability**—all of which we find to be critical in sparse-input scenarios.
>
> We will revise the related work section to better clarify these distinctions and ensure our contributions are clearly stated in the final version.
>
> **Q2:** "More baseline comparisons: The paper claims state-of-the-art performance but omits recent advances such as those referenced in [1]-[6]. Comparing against the latest methods is essential to validate superiority. Additionally, since the approach relies on view interpolation, comparisons with two-view reconstruction methods (e.g., pixelSplat [7], MVSplat [8], NoPosSplat [9]) are suggested to be included into the experiment."
>
> **A2:**  Thank you for this valuable suggestion. We are excited to share new experimental results added during the rebuttal, which demonstrate that our method consistently outperforms recent training-based baselines on both seen and unseen datasets— **despite operating in a zero-shot setting with no fine-tuning or task-specific training**.
>
> While our primary focus is on enhancing 3D Gaussian Splatting (3D-GS) under sparse-input settings with zero-shot diffusion priors, rather than directly generating scenes via feed-forward diffusion (as clarified in response **A1**), we agree that additional comparisons can further contextualize our contributions.
>
> We conducted further experiments against representative feed-forward baselines, including **ReconX, ViewCrafter, and MVSplat360**. Due to time and resource constraints, we focused on methods with public access or reported benchmarks.   ReconX and ViewCrafter as two typical video-generation works for comparison, whereas MVSplat 360 (MVSplat360: Feed-Forward 360 Scene Synthesis from Sparse Views) is a representative feed-forward 3DGS work, one of the latest works in this domain, and it beats pixelSplat [7] and MVSplat [8].  Other concurrent CVPR2025 papers either have not released their code or have adopted different evaluation criteria from ours such that we are unable to fairly reproduce or compare against them due to time constraints and limited accessibility during the rebuttal phase.  We will try to include further comparisons in our final version.
>
>  *Table R4-1: Comparing our zero-shot method with the recent generative-based methods on the  DL3DV dataset.*
>
> |  |  | 3 Views |  |  | 6 Views |  |  | 9 Views |  |
> | --- | --- | --- | --- | --- | --- | --- | --- | --- | --- |
> |  | PSNR↑ | SSIM↑ | LPIPS↓ | PSNR↑ | SSIM↑ | LPIPS↓ | PSNR↑ | SSIM↑ | LPIPS↓ |
> | ReconX | **14.73** | 0.411 | **0.453** | 17.23 | 0.466 | 0.429  | 18.63 |  0.591  | 0.377 |
> | ViewCrafter | 13.45 | 0.402 | 0.473 | 16.22 | 0.421 | 0.435 | 18.23 | 0.553 | 0.348 |
> | Ours | 14.62 | **0.471** | 0.491 | **17.35** | **0.566** | **0.396** | **19.16** | **0.631** | **0.341** |
>
> Despite relying on **frozen diffusion models and no training**, our method delivers comparable or stronger performance than methods specifically trained on DL3DV.   ReconX results are taken from their paper; ViewCrafter and ours were evaluated directly.
>
> *Note: As ReconX has not released the code and model, we report their performance on the DL3DV dataset based on their paper. Besides, as the training data of MVSplat-360 includes our test set on DL3DV, and retraining their model exceeds our available computational resources, we did not test its performance on DL3DV, and we chose to use the MipNeRF-360 dataset for fair benchmarking (discussed below).*
>
> *Table R4-2 Generalization comparison with baselines on MipNeRF-360 datasets. We follow our DL3DV convention to select 9 sparse views for each scene.*
>
> |  | PSNR↑ | SSIM↑ | LPIPS↓ |
> | --- | --- | --- | --- |
> | MVSplat 360 | 14.86 | 0.321 | 0.528 |
> | ViewCrafter | 15.68 | 0.382 | 0.551 |
> | Ours | **17.60** | **0.469** | **0.405** |
>
> On this unseen benchmark—unseen by all methods—our framework generalizes significantly better. Training-heavy baselines show color drift and content inconsistencies, while our method remains robust under sparse-view input.
>
> Key observations:
>
> - **Generalization gap**: ViewCrafter performs well on DL3DV (its training domain) but degrades on MipNeRF-360. Our zero-shot pipeline maintains strong performance across both.
>
> - **Resolution limitations**: MVSplat360 (480p) and ViewCrafter (320×512) are constrained by memory usage. In contrast, our pipeline produces high-resolution outputs (576×1024), consistent with the SVD backbone.
>
> - **Practicality**: Training-heavy methods like See3D require **~25 days on 114×A100 GPUs**, and often **take minutes to render a single view**—limiting their usability in real-time or interactive settings. Our method supports **real-time rendering (~400 FPS)**, making it much more suitable for practical deployment.
>
> In summary, while feed-forward baselines show impressive progress, our method offers a compelling alternative: it leverages frozen generative priors and efficient 3D representations to enable high-quality, real-time novel view synthesis without retraining. We hope this inspires further integration between generative models and structured 3D representations.
>
> We will update the paper to include these new results and clarify the distinctions in methodology, efficiency, and deployment.
>
> **Q3:** "Missing input condition visualization: Visualization experiments in the main paper lack explicit input sparse views. Including visualization of the sparse input images would better illustrate the method’s interpolation capabilities."
>
> **A3:** Thank you for pointing this out.  Showing the sparse input views alongside the synthesized outputs will help better illustrate the interpolation behavior of our method. In the revised version, we will include the original input images in the main qualitative figures to improve clarity and support visual interpretation.

---

> > ### Comment · Reviewer_pPRG · 2025-08-04
> >
> > The rebuttal well addressed my concerns on the comparisons with recent works. The authors are encouraged to demonstrate this clearly in the final version. I will raise my final score.

---

> > > ### Author Response · Authors · 2025-08-05
> > >
> > > Thank you for your thoughtful reviews and for raising the score.
> > > We are glad the rebuttal helped address your concerns, and we’ll make sure the final version clearly highlights the updated comparisons as suggested.

---

### Official Review · Reviewer_xfoR · 2025-07-01

**Clarity:** 4
**Significance:** 2
**Originality:** 2
**Rating:** 4
**Confidence:** 4

**Summary:**

- This paper reframes sparse-input novel view synthesis as a natural video completion task and introduces a zero-shot, generation-guided reconstruction pipeline that integrates pretrained video diffusion models with 3D Gaussian Splatting (3D-GS).

- The method leverages uncertainty-aware modulation to generate pseudo views at intermediate camera poses, which are used to densify and refine the 3D representation. Iterative refinement and primitive densification significantly improve rendering quality, particularly in under-observed regions.

- Experiments on DL3DV, LLFF, and DTU datasets demonstrate strong performance improvements over baseline 3D-GS methods.

**Questions:**

Mentioned in the weaknesses

**Ethical Concerns:**

["NO or VERY MINOR ethics concerns only"]

**Final Justification:**

Thank you for your detailed rebuttal! I will maintain my current rating!

**Limitations:**

Mentioned in the weaknesses

**Paper Formatting Concerns:**

No concerns

**Quality:**

3

**Strengths And Weaknesses:**

**Strengths**
- The reinterpretation of novel view synthesis as a video frame completion task is conceptually elegant and allows the use of powerful pretrained video diffusion models. The use of best/second-best/third-best rankings in quantitative evaluation adds credibility and fairness to the reported improvements. (Table 1, Tabel 2)

- Supplementary videos show strong qualitative performance in scene reconstruction, particularly in maintaining structural coherence and reducing rendering artifacts. (video.mp4 01:16-02:21)

- The proposed uncertainty-aware modulation and primitive densification modules demonstrate a well-designed synergy between generative priors and 3D representation learning. (Page 2 - Introduction)

**Weaknesses**

**Major**
- Despite improvements, the method still shows limitations in rendering fine-grained details, especially in texture-rich regions. For example, like artifacts in the video. (like 01:37 in the supplementary video, GT-Ours)

- The paper claims a significant reduction in rendering artifacts, but visual results—while improved—still show occasional imperfections, which should be acknowledged more explicitly. (like 01:42 / 02:01 in the supplementary video, GT-Ours)

**Minor**
- While the provided supplementary videos are promising, the number of qualitative examples (Both paper and supplementary video showing 5 examples) looks insufficient to fully validate generalizability. This is non-critical, but adding more diverse scenes would strengthen the experimental evidence.

---

> ### Author Rebuttal · Authors · 2025-07-30
>
> # Reviewer xfoR
>
> **Q1:** "Despite **improvements**, the method still shows limitations in rendering fine-grained details, especially in texture-rich regions. For example, like artifacts in the video. (like 01:37 in the supplementary video, GT-Ours)"
>
> **A1:** We agree that some loss of fine-grained detail—such as the artifact at 01:37 in the supplementary video (GT vs. Ours)—can occur in texture-rich or underconstrained regions. This is a known challenge in sparse-input, wide-baseline settings, where details like dense foliage or fine textures are not fully supported by geometry alone. Similar limitations have been reported in prior works such as DNGaussian and 3DGS-Enhancer.
>
> Most importantly, our method operates in a **fully zero-shot** setting, using only a few posed input images and a **frozen** diffusion model, without any task- or scene-specific fine-tuning. Despite this, it achieves globally consistent geometry and appearance across most views, as demonstrated throughout the supplementary video.
>
> To mitigate detail loss, we introduce two key mechanisms:
> - Uncertainty-aware modulation (Sec. 4.1), which stabilizes pseudo-view generation, and
> - Perceptual LPIPS loss (Eq. 6), which improves supervision signal beyond pixel alignment.
>
> These design choices lead to measurable and visible improvements in reconstruction quality, as shown in Table 3 and the qualitative comparisons.
>
> That said, we recognize that further improvement in high-frequency texture recovery remains a valuable direction. Future extensions could incorporate semantic priors, generalizable refinement networks, or lightweight super-resolution modules to better capture fine details under limited-view conditions.
>
> **Q2:** "The paper claims a significant reduction in rendering artifacts, but visual results—while improved—still show **occasional** **imperfections**, which should be acknowledged more explicitly. (like 01:42 / 02:01 in the supplementary video, GT-Ours)"
>
> **A2:** We thank the reviewer for highlighting these examples. We agree that occasional imperfections—such as those at 01:42 /  02:01 in the supplementary video—can appear, particularly in extreme viewpoints or complex scenes. These limitations stem from the generative bounds of the frozen video diffusion model and the inherent under-constraint in sparse-view inputs.
>
> That said, our method operates under **strict zero-shot** conditions, using **no task- or scene-specific fine-tuning**, which distinguishes it from most prior approaches. Despite these constraints, our pipeline achieves **strong qualitative and quantitative performance**, with consistent geometry and appearance across views.
>
> To address these challenges, we introduce two targeted strategies:
> - **Uncertainty-aware modulation**, which stabilizes pseudo-view generation, and
> - **Perceptual supervision via LPIPS**, which improves robustness to generative artifacts.
>
> These components contribute significantly to performance improvements, as shown in our ablation: PSNR improves from 16.59 (baseline 3D-GS) to 19.16 with our full method (Table 3).
>
> We will revise the manuscript to explicitly acknowledge these remaining imperfections and position them as valuable opportunities for future enhancement—e.g., through lightweight refinement or adaptive guidance methods—while emphasizing the strength of our results given the zero-shot setting.
>
> **Q3:** "While the provided supplementary videos are promising, the number of qualitative examples (Both paper and supplementary video showing 5 examples) looks insufficient to fully validate generalizability. This is non-critical, but adding more diverse scenes would strengthen the experimental evidence."
>
> **A3:** We thank the reviewer for the helpful suggestion. Due to submission file size constraints, we focused the supplementary video and paper on a concise set of representative scenes -- **8 scenes in DL3DV, 3 scenes in DTU**, and **3 scenes in LLFF** -- chosen to reflect diverse geometry, texture, and baseline conditions.
>
> Importantly, our **quantitative evaluation is conducted over the full test sets of each dataset**, ensuring statistically meaningful assessment of generalizability. The qualitative examples are intended as illustrative highlights rather than comprehensive coverage.
>
> We will expand the qualitative results in the final version of the paper and supplementary materials to include a broader range of scenes and viewpoints, further supporting the generalization claims.

---

> > ### Comment · Reviewer_xfoR · 2025-08-01
> >
> > Thank you for your detailed rebuttal! I will maintain my current rating!

---

> > > ### Author Response · Authors · 2025-08-03
> > > **Thank You for Your Positive Rating**
> > >
> > > Dear Reviewer xfoR,
> > >
> > > Thank you for your positive final rating. We’re glad to know that our rebuttal helped clarify our contributions and address your concerns.
> > >
> > > We truly appreciate your thoughtful review and the time you dedicated to our submission.
> > >
> > > Best regards,
> > >
> > > The Authors of Submission 4240

---

### Official Review · Reviewer_axbY · 2025-07-01

**Clarity:** 3
**Significance:** 2
**Originality:** 3
**Rating:** 5
**Confidence:** 5

**Summary:**

This paper presents a pipeline for solving sparse-view novel view synthesis (NVS) by leveraging a pretrained stable video diffusion model. Given a guidance image and an uncertainty mask, the model generates pseudo-views, which are then used to iteratively refine a 3D Gaussian Splatting (3DGS) reconstruction, leading to high-quality final results.

**Questions:**

Please refer to the points in the weakness.

**Ethical Concerns:**

["NO or VERY MINOR ethics concerns only"]

**Final Justification:**

I am impressed by the strong performance compared to baseline methods. Rather than following the popular data-driven trend, the proposed approach demonstrates that pretrained models can be highly effective when used with careful design and insight.

**Limitations:**

Yes

**Paper Formatting Concerns:**

No obvious formatting issues.

**Quality:**

3

**Strengths And Weaknesses:**

**Strengths:**
1. Directly using a pretrained diffusion model without finetuning is insightful in terms of computational resources.

**Weaknesses:**
1. The MipNeRF-360 dataset, which is one of the most challenging benchmarks for outdoor unbounded NVS, is not evaluated in this paper.
2. Strong baselines that also utilize video diffusion priors, such as **3DGS-Enhancer** [1] and **MVSplat360** [2], are missing from the comparison.
3. The motivation for using a pretrained video diffusion model is questionable. Since SVD includes generative priors from dynamic and synthetic content, this may conflict with the goal of optimizing a static 3D representation. Although finetuning requires significant computational resources, it can substantially improve performance.
4. Previous works [1][2] highlight the degradation caused by the VAE component. Without aligning the latent space between the rendered pseudo-views and the ground truth images, quality issues such as color shifts and distortions can arise. This paper does not address this critical point.
5. Based on both image and video visualizations, noticeable inconsistencies, artifacts, and color shifts are observed. While directly using SVD without finetuning significantly reduces computation, it also inevitably sacrifices performance, particularly because enhancements such as those discussed in points 3 and 4 cannot be incorporated.

**References:**
[1] *3DGS-Enhancer: Enhancing Unbounded 3D Gaussian Splatting with View-consistent 2D Diffusion Priors*
[2] *MVSplat360: Feed-Forward 360 Scene Synthesis from Sparse Views*

---

> ### Author Rebuttal · Authors · 2025-07-31
>
> **Q1+Q2:** "The MipNeRF-360 dataset, which is one of the most challenging benchmarks for outdoor unbounded NVS, is not evaluated in this paper." and "Strong baselines that also utilize video diffusion priors, such as 3DGS-Enhancer [1] and MVSplat360 [2], are missing from the comparison."
>
> **A1+A2:** Thank you for the helpful suggestions. We have extended our experiments to include comparisons with 3DGS-Enhancer and MVSplat360, and provided additional benchmarking results on the challenging MipNeRF-360 dataset.  These results demonstrate that our method exhibits **strong generalization ability** across diverse and unseen scenarios.
>
> As a lightweight **zero-shot** approach, our method **significantly outperforms** both **3DGS-Enhancer** and **MVSplat360** on the MipNeRF-360 benchmark, despite requiring **no diffusion model training or task-specific finetuning**. On the DL3DV dataset, where **3DGS-Enhancer** was specifically trained, our method still achieves **comparable or even superior performance**.
>
> The detailed quantitative comparisons are provided below.
>
> (1) **Comparison with 3DGS-Enhancer and DNGaussian (a representative method on sparse-input 3DGS) on DL3DV**
>
> *Table R2-1* shows that our method achieves **comparable or even superior performance** to 3DGS-Enhancer, despite operating in a **zero-shot** manner. In contrast, 3DGS-Enhancer requires **GPU-intensive training** of the diffusion model specifically on the DL3DV dataset.
>
> Note that we do not include **MVSplat360** for fairness, as its training set includes our test set on DL3DV.
>
> *Table R2-1: Results on the DL3DV dataset.* We hold every eighth image as our test split, and
> evenly sample 9 sparse views from the remaining images for training the 3D-GS. (Please refer to the reviewer sLmb's section for more results.)
>
> |  | PSNR↑ | SSIM↑ | LPIPS↓ |
> | --- | --- | --- | --- |
> | DNGaussian | 13.44  | 0.365 |  0.539 |
> | 3DGSEnhancer | 18.50 | 0.630 | 0.305 |
> | Ours | 19.16 | 0.631 | 0.341 |
>
> (2) **Comparisons on MipNeRF-360 Dataset.**
>
> We further evaluate our method on the challenging MipNeRF-360 dataset, comparing it against three competitive baselines.
>
> *Table R2-2 Results on MipNeRF-360. We hold every eighth image as our test split, and evenly sample 9 sparse views from the remaining for 3D-GS optimization.*
>
> |  | PSNR↑ | SSIM↑ | LPIPS↓ |
> | --- | --- | --- | --- |
> | DNGaussian | 12.51  | 0.228  | 0.683 |
> | 3DGS-Enhancer | 16.22 | 0.399 | 0.454 |
> | MVSplat 360 | 14.86  | 0.321 | 0.528 |
> | Ours | **17.60** | **0.469** | **0.405** |
>
> *Table R2-2* shows that our method **outperforms all baselines across all evaluation metrics**, demonstrating strong generalization and robustness on unseen outdoor scenes. We found that although 3DGS-Enhancer (specifically trained on **DL3DV** dataset) performs comparably to ours on DL3DV (see Table R2-1),  it exhibits significant **performance degradation** on MipNeRF-360, suggesting limited cross-dataset generalization. Similarly, **MVSplat360 suffers on this dataset, where its outputs exhibit notable color shifts and semantic inconsistencies.** We will include qualitative comparisons in the final version of the paper to illustrate these issues.
>
> Additionally, the feed-forward method MVSplat360 is constrained by computational and memory overhead, and its publicly released model is limited to generating 480p images. In contrast, our method maintains high-resolution synthesis capability while requiring no additional diffusion model training or task-specific finetuning, which is more scalable and efficient for real-world deployment.
>
> We will update the main paper and supplementary material to include this evaluation and comparative analysis.
>
> **Q3:** "The motivation for using a pretrained video diffusion model is questionable. Since SVD includes generative priors from dynamic and synthetic content, this may conflict with the goal of optimizing a static 3D representation. Though finetuning requires significant computational resources, it can substantially improve performance."
>
> **A3:** Thank you for the insightful comment. While it is true that SVD's training data includes dynamic content, we find its inductive biases generalize well to static camera transitions, especially when guided by our structural constraints.
>
> 1. SVD captures strong spatiotemporal priors that promote coherent view transitions and occlusion-aware reasoning, likely due to its text-to-image pretraining.
> 2. Our **uncertainty-aware modulation** enforces geometric and photometric consistency, reducing hallucinations and improving alignment, as demonstrated in Table 3(a).
>
> The supplementary video (2:10–2:20) also shows that the generated views maintain static scene properties, with minimal negative impact on 3D-GS results.
>
> In summary, while fine-tuning could improve task-specific alignment, our results show that, **with proper guidance**, a pretrained SVD model can already serve as a **practical and effective prior** for static 3D-GS—supporting **lightweight, zero-shot deployment** in resource-constrained settings.
>
> **Q4:** "Previous works [1,2] highlight the degradation caused by the VAE component. Without aligning the latent space between the rendered pseudo-views and the ground truth images, quality issues such as color shifts and distortions can arise. This paper does not address this critical point.”
>
> **A4:** We thank the reviewer for raising this important point. We fully agree that VAE-induced degradation is a known issue, and we have made a concerted effort to address it through a set of lightweight but effective mechanisms—designed specifically to preserve our zero-shot setting without retraining the diffusion model. While these strategies are already part of our current method, we appreciate the opportunity to better highlight their role in the paper and will revise the manuscript to clarify this contribution.
>
> While prior works such as 3DGS-Enhancer [1] and MVSplat360 [2] address this through **explicit training** of the diffusion model to better align the latent space, our work **intentionally avoids diffusion model retraining** to preserve our **zero-shot design**. Instead, we propose specific mechanisms to **robustify the 3D-GS optimization under these non-ideal generative conditions and alleviate color drifts and distortion via the underlying 3D-GS representation:**
>
> - **Perceptual Loss for Robust Supervision:** We replace the standard L1 loss with the **LPIPS loss** in Eq. (6), which encourages perceptual similarity rather than strict pixel-wise alignment. This makes the 3D-GS training **less sensitive to moderate color shifts or spatial inconsistencies** in the generated pseudo views. The effectiveness is confirmed by our ablation in Table 3(b) and the qualitative improvements in the supplementary video.
> - **Feedback via Uncertainty-Aware Modulation:**  Our framework incorporates an **uncertainty-aware modulation mechanism**, which uses the intermediate 3D-GS representation to guide the diffusion model during pseudo-view synthesis. Since the 3D-GS maintains a coherent, geometry-aware representation, this feedback helps suppress inconsistencies and stabilizes the appearance across views.
> - **Latent Space Scaling:** Empirically, we observed that **scaling the conditional latent variables** from the pseudo views (i.e., VAE latents) **helps mitigate color saturation and drift**, likely by regularizing the decoder’s sampling behavior.  While not the core focus of our technical contribution, this adjustment improved the visual consistency and will be more clearly described in the final version.
>
> The 3D-GS renderings (1:17–2:03) in the supplementary video show limited color drifts and distortions due to our mechanisms.
>
> In summary, while we do not directly modify the diffusion model, our method introduces **lightweight yet effective solutions** for mitigating VAE-induced degradation, all while maintaining the advantages of **zero-shot inference and generalizability**.
>
> **Q5:** "Based on both image and video visualizations, noticeable inconsistencies, artifacts, and color shifts are observed. While directly using SVD without fine-tuning **significantly reduces computation**, it also inevitably sacrifices performance, particularly because enhancements such as those discussed in points 3 and 4 cannot be incorporated."
>
> **A5:** We appreciate the reviewer’s observation and agree that pseudo-views generated by SVD may exhibit minor artifacts such as color shifts or local inconsistencies, especially in wide-baseline or occluded regions. However, our approach is deliberately designed to address this challenge through lightweight, principled mechanisms that preserve the zero-shot nature of the pipeline—without resorting to costly diffusion model fine-tuning.
>
> Our framework prioritizes generalization, efficiency, and deployment practicality. While some prior methods mitigate such artifacts through scene-specific fine-tuning (e.g., 3DGS-Enhancer [1]), they incur significant computational overhead and require paired supervision or GPU access—making them less viable for real-world applications.
>
> In contrast:
>
> - **In resource-constrained or edge environments**, retraining large generative models is infeasible. Our inference-only pipeline allows for fast, GPU-friendly execution.
> - **In field scenarios** such as surveying, architecture, or cultural preservation, our method supports quick adaptation to novel scenes without retraining or manual intervention.
> - **Compared to training-heavy approaches** like MVSplat360, which face generalization limitations and are restricted to low-resolution outputs (e.g., 480p), our method supports high-resolution, cross-dataset reconstructions (see A1/A2).
>
> This trade-off is intentional: our method avoids retraining while still delivering high-resolution 3D reconstructions with reliable consistency and quality across diverse scenes. We will revise the paper to better highlight this design philosophy and its practical benefits.

---

> > ### Comment · Reviewer_axbY · 2025-08-04
> >
> > Thank you for your detailed rebuttal.
> >
> > I am impressed by the strong performance compared to baseline methods.
> > Rather than following the popular data-driven trend, the proposed approach demonstrates that pretrained models can be highly effective when used with careful design and insight.
> > Moreover, I encourage the authors to include more visualizations in the final version.
> >
> > I have updated my score to 5.

---

> > > ### Author Response · Authors · 2025-08-04
> > >
> > > Dear Reviewer axbY,
> > >
> > > Thank you for your thoughtful feedback and for increasing your rating—we sincerely appreciate your support.
> > >
> > > We’re glad our approach and design choices resonated with you, and we’ll be sure to include additional visualizations in the final version as you suggested.
> > >
> > > Best regards,
> > >
> > > The Authors of Submission4240

---

### Official Review · Reviewer_sLmb · 2025-07-02

**Clarity:** 4
**Significance:** 3
**Originality:** 4
**Rating:** 5
**Confidence:** 3

**Summary:**

This paper addresses the challenging problem of novel view synthesis from sparse input images, specifically within the 3D Gaussian Splatting (3DGS) framework. The authors reframe this task as a video completion problem, proposing a novel generation-guided pipeline that leverages a pretrained video diffusion model to fill in the "gaps" between sparse views. The core of the method is an iterative process. It begins with an initial 3DGS model. This model is used to generate depth maps, which in turn are used to create "guidance images" by warping pixels from the nearest real input views. Crucially, the paper introduces an uncertainty-aware modulation mechanism; it evaluates the reliability of these guidance images via a cross-view consistency check and uses this uncertainty to steer the video diffusion model. The diffusion model is tasked with generating plausible content primarily in high-uncertainty (e.g., occluded or under-observed) regions while preserving reliable content from the guidance images. These generated "pseudo-views" are then used as additional supervision to refine and densify the 3DGS representation. This cycle is repeated to progressively improve the final 3D scene. The authors demonstrate great performance on several challenging datasets, most notably DL3DV.

**Questions:**

1. Are there any failure cases related to the weaknesses discussed in point 1 (Critical reliance on SVD)? Specifically, what happens if the SVD generates undesirable hallucinations that are both appearance- and geometry-inconsistent? If so, are there any proposed solutions to address this issue?

2. Can this proposed method outperform other generative-based methods with similar technical approaches? Such as Viewcrafter [1] (feed-back loop between generated views and 3D representation) and 3DGS-enhancer [2] (frames interpolation by video diffusion).

[1] Yu, Wangbo, et al. "Viewcrafter: Taming video diffusion models for high-fidelity novel view synthesis." Open Source Project 2024.

[2] Liu, Xi, Chaoyi Zhou, and Siyu Huang. "3dgs-enhancer: Enhancing unbounded 3d gaussian splatting with view-consistent 2d diffusion priors." NeurIPS 2024.

Justification of the rating:
I think overall, this paper presents a technically novel approach with good performance, with clear writing and organization. However, as a generative-based method, comparisons with other generative-based baselines are necessary. Therefore, my rating is borderline accept because of limited evaluation.

**Ethical Concerns:**

["NO or VERY MINOR ethics concerns only"]

**Final Justification:**

I think overall, this paper presents a technically novel approach with good performance, with clear writing and organization. However, as a generative-based method, comparisons with other generative-based baselines are necessary. Therefore, my intial rating is borderline accept because of limited evaluation. As the authors have resolved my concerns on evaluation and showcased the effectiveness of the proposed method in their rebuttal, I would like to increase my rating to accept.

**Quality:**

3

**Strengths And Weaknesses:**

Strengths:

1. The paper is well-written and easy to follow, the writing quality is great.

2. Zero-shot is great. While prior works have used diffusion models, they have largely focused on image-based priors or required fine-tuning. This work's use of a zero-shot, pre-trained video diffusion model is a significant and novel contribution, as it inherently models the spatio-temporal consistency between views. This approach provides a new and effective direction for tackling the ill-posed nature of sparse-view reconstruction.

3. The technical execution of the core idea is excellent. The proposed pipeline is well-designed and sophisticated. The "uncertainty-aware modulation" is a particularly clever mechanism. Instead of naively using generated views, it provides a principled way to fuse geometric information from the current 3D-GS reconstruction with the powerful generative prior of the diffusion model.

Weaknesses:

1. (major weakness) Missing comparisons with other generative-based methods: The paper only compares against non-generative sparse-view NeRF or 3DGS approaches, but does not include any baselines that also leverage generative priors. Such comparisons are important for a fair evaluation.

2. (minor weakness) Critical reliance on SVD: The proposed method heavily relies on the capabilities of the underlying video diffusion model (SVD), as the authors acknowledge. As a result, it may struggle with out-of-distribution scenes, where the generative prior can hallucinate plausible but inaccurate details (appearance- and geometry-inconsistent contents), potentially corrupting the resulting 3DGS.

---

> ### Author Rebuttal · Authors · 2025-07-30
>
> # Reviewer sLmb
>
> **Q1:** "(major weakness) Missing comparisons with other generative-based methods: The paper only compares against non-generative sparse-view NeRF or 3DGS approaches, but does not include any baselines that also leverage generative priors. Such comparisons are important for a fair evaluation." and "Can this proposed method outperform other generative-based methods with similar technical approaches? Such as Viewcrafter [1] (feed-back loop between generated views and 3D representation) and 3DGS-enhancer [2] (frames interpolation by video diffusion)."
>
> **A1:** We thank the reviewer for highlighting the importance of comparing against generative-based methods.
> In our original submission, we focused on zero-shot NeRF and 3D-GS baselines, which, like our approach, do not require computation-intensive training or fine-tuning of the diffusion models. This setup was chosen to emphasize practical deployment scenarios where compute resources and data collection are limited.
>
> To directly address the reviewer’s concern, we have now included additional comparisons with ViewCrafter [1] and 3DGS-Enhancer [2] on the DL3DV and MipNeRF-360 datasets. These results are provided below. Note that the training set of the released ViewCrafter model includes our test set on DL3DV, and we present the results here for reference only.
>
> *Table R1-1: Comparing our zero-shot (training-free) method with the recent generative-based methods (that need expensive GPU hours for diffusion training) on the **DL3DV** dataset. Our method achieves comparable or even better performance.*
>
> |  |  | 3 Views |  |  | 6 Views |  |  | 9 Views |  |
> | --- | --- | --- | --- | --- | --- | --- | --- | --- | --- |
> |  | PSNR↑ | SSIM↑ | LPIPS↓ | PSNR↑ | SSIM↑ | LPIPS↓ | PSNR↑ | SSIM↑ | LPIPS↓ |
> | ViewCrafter*  | 13.45 | 0.402 | 0.473 | 16.22 | 0.421 | 0.435 | 18.23 | 0.553 | 0.348 |
> | 3DGSEnhancer | 14.33  | 0.424 | **0.464** | 16.94 | 0.565 | **0.356**  | 18.50 | 0.630 | **0.305** |
> | Ours | **14.62** | **0.471**  | 0.491 | **17.35** | **0.566** | 0.396 | **19.16**  | **0.631** | 0.341 |
>
>
> *Table R1-2 Comparison with baselines on **MipNeRF-360** datasets. We follow the convention on DL3DV  to select 9 sparse views for each scene.*
>
> |  | PSNR↑ | SSIM↑ | LPIPS↓ |
> | --- | --- | --- | --- |
> | ViewCrafter | 15.68 | 0.382  | 0.551 |
> | 3DGSEnhancer | 16.22 | 0.399 | 0.454 |
> | Ours | **17.60** | **0.469** | **0.405** |
>
> The results show that:
>
> - Our method achieves comparable or superior performance to both ViewCrafter and 3DGS-Enhancer, despite being significantly more lightweight and requiring no additional training of the diffusion model.
> - While 3DGS-Enhancer (trained on DL3DV) performs slightly better on DL3DV in some metrics, we observe notable generalization issues when tested on the unseen MipNeRF-360 dataset, where our zero-shot method consistently outperforms it. A similar degradation has also been observed for ViewCrafter.
>
> **Q2** "(minor weakness) Critical reliance on SVD: The proposed method heavily relies on the capabilities of the underlying video diffusion model (SVD), as the authors acknowledge. As a result, it may struggle with out-of-distribution scenes, where the generative prior can hallucinate plausible but inaccurate details (appearance- and geometry-inconsistent contents), potentially corrupting the resulting 3DGS." and "Are there any failure cases related to the weaknesses discussed in point 1 (Critical reliance on SVD)? Specifically, what happens if the SVD generates undesirable hallucinations that are both appearance-and geometry-inconsistent? If so, are there any proposed solutions to address this issue?”
>
> **A2:** Thank you for your insightful comments. We agree that our method relies on the generative capability of the underlying video diffusion model (SVD), and appreciate the opportunity to clarify how we address potential failure cases and hallucinations.
>
> In our experiments, we primarily observed two types of inconsistencies:
>
> 1. Moderate temporal inconsistency in distant regions with weak geometric support or fine-grained textures (e.g., grass, tree leaves), where the diffusion model may generate visually plausible but slightly inconsistent content.
> 2. Occasional generation collapse in individual frames, particularly in early iterations when the 3D-GS guidance is still inaccurate or the input views are extremely sparse.
>
> To mitigate **Type 1**, as described in Lines 236–238, we replace L1 loss with LPIPS loss during 3D-GS supervision using pseudo-views. This encourages perceptual-level alignment and helps reduce the impact of subtle inconsistencies. The effectiveness of this strategy is demonstrated in Table 3(b) in our paper.
>
> For **Type 2**, while we acknowledge that hallucinations can occasionally appear, we find that they do not significantly affect final 3DGS convergence. These artifacts are typically isolated, non-persistent, and often suppressed by the global optimization process of 3D-GS, which relies on multi-view consistency.
>
> That said, we recognize that such hallucinations can degrade reconstruction quality. We consider these unreliable generations as transient distractors, and future work can incorporate recent advances such as “SpotLessSplats: Ignoring Distractors in 3D Gaussian Splatting”, which explicitly addresses the suppression of distractors in 3D Gaussian Splatting. Integrating such filtering mechanisms could further improve the robustness of our pipeline.
>
> Finally, while the capabilities of our method are inherently bounded by the current performance of SVD, we are optimistic that diffusion models will continue to improve significantly with scale. Our pipeline is designed to be modular and compatible with future generative backbones, making it well-positioned to benefit from the ongoing progress in video generation.

---

> > ### Comment · Reviewer_sLmb · 2025-08-06
> > **Reply to the authors**
> >
> > Thank you for the detailed response — I believe my concerns have been fully addressed. Given the clear demonstration of both novelty and effectiveness, I would like to increase my rating to 5.

---

> > > ### Author Response · Authors · 2025-08-07
> > >
> > > Thank you again for your valuable comments. We truly appreciate your constructive feedback and support!

---

### Note · Authors · 2025-08-13

We sincerely thank all reviewers for their constructive feedback and engagement. We are pleased that our rebuttal, including new experiments on MipNeRF-360 and comparisons with generative baselines (ViewCrafter, 3DGS-Enhancer, MVSplat360), successfully addressed their main concerns, leading three reviewers to raise their scores.

Our work introduces a novel, zero-shot framework that effectively leverages pretrained video diffusion models for sparse-input novel view synthesis. The experimental results confirm its strong performance and superior generalization compared to training-heavy methods, highlighting its practical value.

We will integrate all promised additions, including the new baseline comparisons, expanded evaluations, and additional visualizations, into the final paper to fully reflect the outcomes of this discussion. We thank the reviewers and the AC for their time and believe our paper will be a valuable addition to the conference.

---

### Decision · Program_Chairs · 2025-09-17

**Decision:**

Accept (poster)

**Comment:**

This work makes a valuable contribution to sparse-view novel view synthesis by demonstrating that carefully designed zero-shot approaches can compete with training-intensive methods while offering superior generalization and deployment advantages. The technical innovation, experimental validation, and practical significance justify acceptance. The consistent positive reviewer feedback, successful resolution of all major concerns, and great experimental results across diverse datasets demonstrate the work's quality and impact potential.